# Analysing Power Relations among Older Norwegian Patients and Spanish Migrant Nurses in Home Nursing Care: A Critical Discourse Analysis Approach from a Transcultural Perspective

**DOI:** 10.3390/healthcare11091282

**Published:** 2023-04-29

**Authors:** Pablo Martínez-Angulo, Manuel Rich-Ruiz, Pedro E. Ventura-Puertos, Salvador López-Quero

**Affiliations:** 1Department of Nursing, Pharmacology, and Physiotherapy, Faculty of Medicine and Nursing, University of Córdoba (UCO), 14004 Córdoba, Spain; n22maanp@uco.es (P.M.-A.);; 2Interdisciplinary Research Group in Discourse Analysis (HUM380), University of Córdoba (UCO), 14071 Córdoba, Spain; 3Maimonides Biomedical Research Institute of Córdoba (IMIBIC), Hospital Universitario Reina Sofía (HURS), 14004 Córdoba, Spain; 4Ciber Fragility and Healthy Aging (CIBERFES), 28029 Madrid, Spain; 5Nursing and Healthcare Research Unit (Investén-isciii), Instituto de Salud Carlos III, 28029 Madrid, Spain; 6Department of Language Sciences, Faculty of Philosophy and Letters, University of Córdoba (UCO), 14003 Córdoba, Spain

**Keywords:** older Norwegian patients, Spanish migrant nurses, nurse migration, active listening, shared decision-making, patient participation, transcultural nursing, cultural competency, critical discourse analysis

## Abstract

Power relations in care are the link between patients and nurses regarding communication and the ability to act in this context. It can be affected when there is cultural interference between members, putting mutual understanding at risk in healthcare situations. This study analyses power relations in healthcare situations between older Norwegian patients and Spanish migrant nurses regarding active listening, shared decision-making, and patient participation. We performed a hermeneutical study endorsed in critical discourse studies framework from a transcultural perspective. A purposive sampling included older Norwegian patients living alone and Spanish migrant nurses working in Norway. Eleven face-to-face semi-structured interviews were conducted with older patients and four via videoconference with migrant nurses. The analysis followed hermeneutic considerations by Crist and Tanner, and linguistic analysis was performed. Shared decision-making and active listening situations sometimes showed a power imbalance that negatively influenced older Norwegian patients. However, Spanish migrant nurses were also conditioned by care organising institutions. This power triangle negatively affected the relationship between the older patients and migrant nurses, resulting in a lack of communication, personnel, time and trust. The migratory experience influenced the care provided by Spanish migrant nurses, shaping a series of cultural competencies acquired through the migratory process.

## 1. Introduction

Nursing is the most significant health professional group in the world; that is why nurses represent a crucial figure in providing quality and good care, from a socioeconomic point of view, in a globalised and industrialised society where needs urgently have to be satisfied [1]. According to the state labour market report for 2021 in Spain, nursing was one of the occupations with the most difficulty filling positions, and whose temporary rate in labour contracts was 92.84%. Of 8836 registered contracts, 8105 were of temporary duration, giving us a sample of the high percentage of instability in Spanish nursing employment [2]. Regarding the type of contract, over the last 10–15 years, it has been observed that the trend in temporary contracts and part-time contracts has been increasing. In the last two years, temporary contracts have maintained very high figures [2]. A recent multicentric cross-sectional study in Spain by Acea-López et al. [3] showed that nurses with a temporary contract suffered from a high burnout rate and high levels of emotional fatigue, depersonalisation and lack of personal fulfilment. Regarding migration, 16.13% of participants considered leaving the country for France, Germany, Ireland, Norway, the United States or the United Kingdom. Therefore, labour conditions, job uncertainty and professional development are central elements in the motivation for Spanish nurses to leap to work abroad [4,5,6], thus establishing Spain as a net supplier of nurses to foreign countries. This is not exceptional in the scientific literature, which has shown that the migration of health professionals in Europe has a long history [7]. However, they are not exempt from fears such as rejection or discrimination when working abroad in the host country [8].

The counterpart in these labour conditions is Norway, which has been experiencing diminishing health workers for years, whose vacancies are sometimes barely filled. Official statistics in Norway suggest that in 2035, there will be a need for 28,000 nurses, further accentuating this coverage deficit [9]. It is then that qualified immigrant caregivers represent an increasingly influential group in mitigating this care deficit [10]. Despite the increasing role and presence of immigrants in health care work throughout Norway, more research is still needed on this group, specifically, those who provide in-home care for older adults [11,12].

In this context, where a situation of cultural vulnerability appears concerning migrant professionals, it is essential to focus these investigations on the relationship of migrant nurses with their professional colleagues and older patients, which may be resented or at least influenced by a changing cultural environment [12,13]. Several studies have been primarily concerned with showing that healthcare practices are influenced by the relationships between migrant caregivers —mainly female—and patients receiving care in domestic and institutional contexts.

In a globalised world where meeting health needs has become a challenge for any nation, it is also essential to explore the ins and outs of the relationships of care in the community, which is why this study is presented as a transcultural window to glimpse the characteristics of these relationships increasingly present in the international reality of care. These relationships include nursing practices and professionals as relational identities strongly influenced by social, cultural, economic and political circumstances [14,15]. These nursing practices play a critical presence in the professional identities of nurses and result in hierarchies, power relations, and empowerment in particular cultural healthcare settings [16,17].

This study focuses on analysing power relations in health care provided by Spanish migrant nurses to older Norwegian people in the community. In our study context, we understand power relations as the phenomenon that modulates the execution of shared decision-making, the expression of preferences and the active participation of older Norwegian patients in the care provided by Spanish migrant nurses. The hierarchical situation both members occupy in this healthcare relationship and the different “ability to act” [18] of both members considering these three components may influence this phenomenon. Addressing the relationship between older people and nurses is, therefore, essential within primary care settings [19]. To this relationship must also be added a robust communicative component, which intervenes in the modulation of the expression of preferences by older patients and in the active listening by migrant nurses, leading to the possible appearance of situations where decision-making takes centre stage [20]. So, the objectives of this study were to analyse power relations regarding situations about active listening to preferences, shared decision-making and participation in care between older Norwegian patients and Spanish migrant nurses in the community; to identify imbalance in the perception of these situations through discourse with older Norwegian patients and Spanish migrant nurses; to explore the effect of culture shock experience on the nursing care through discourse with Spanish migrant nurses; and to highlight the social representations conveyed by older Norwegian patients and Spanish migrant nurses shaped by their discursive strategies.

## 2. Materials and Methods

### 2.1. Design

We conducted a hermeneutic interpretative study endorsed in the critical discourse analysis (CDA) framework of discourse studies (DS) with a transcultural perspective. The hermeneutic design allowed us to discover, through an interpretive process, the meaning of the life experience of the study participants [21]. Language is a powerful tool [22], as is the discourse, understood as a means of reproducing inequality [23]. This inequality can lead to an imbalance of power in the relationships between older Norwegian patients and the Spanish migrant nurses who care for them. In this way, the discourse can influence the appearance and development of the study phenomena. To analyse that potential influence, we adopt the academic-critical perspective of the CDA [24]. We take Van Dijk’s model of power relations as a reference—considering his sociocognitive approach [25]—together with Foucauldian elements such as oppression and society [23,26]. We decided to confer a transcultural perspective to this study through the Campinha–Bacote notions of cultural competence in healthcare delivery [27]. We used this author’s transcultural framework as a reference because we understand cultural competence in nursing as a process, not a consequence. This process encompasses the migrant experience that we bore in our study. In detail, we consider her definition of cultural awareness, cultural skill, cultural encounters and cultural desire as a whole, which cover the process by which nurses attempt to provide quality care while acquiring the ability to work in a culturally different environment and care for a group of patients sharing that foreign cultural context, in other words, to become culturally competent healthcare professionals.

Furthermore, we wish to present this article as a detailed and complete study report. To achieve that, Appendix A contains the 32 items that comprise the COREQ checklist for original qualitative studies [28]. For an adequate presentation of the information on sex throughout the present study, we have followed the considerations reflected in the SAGER guidelines [29].

### 2.2. Setting

Data collection of older Norwegian people was conducted in two different nursing home facilities in Norway. A nursing home facility provides services for the patients who live at home to maintain or regain the ability to perform activities they can master. These services can cover coping with everyday life and rehabilitation, use of welfare technology, help with self-care, medication management and wound treatment [30]. The first of them was in Jessheim. Jessheim is a town in Ullensaker municipality in Viken and had 23,000 inhabitants in 2022 [31]. Jessheim is an administrative centre in Ullensaker municipality in Øvre Romerike. Of the total population in this town, 1189 people were men aged 67 or older, while 1541 were women aged 67 or older [32]. The second was in Våler. Våler is a village in Våler municipality in Innlandet, with 1138 inhabitants in 2022 [33]. Våler is the administrative centre in the Våler municipality. Of the total population in this village, 165 people were men aged 67 or older, while 208 were women aged 67 or older [32].

### 2.3. Study Participants

We performed purposeful sampling according to concrete selection criteria [34], as Table 1 shows. Regarding older Norwegian patients, we decided to use criteria A and B because they are characteristics that intensify the concept of vulnerability in this group and are a reasonably general profile of older patients targeted in nursing home services in Norway. Therefore, it would facilitate the transfer of this context to other studies. Regarding the Spanish migrant nurses, we decided to use criteria F and G because we considered it a reasonable time to determine a varied range of experiences framed in the different phases of culture shock that would provide us with a greater depth of information on the migrant experience in Norway and in-home nursing services.

### 2.4. Recruitment of Participants

Regarding recruiting older Norwegian patients, the first researcher met with the respective nurse supervisors at each nursing home centre to explain the research project and share the selection criteria with them. After this, the nurse supervisors, with the help of the nurses employed at the centres, provided potential older participants with an information booklet containing the fundamental concepts of the project available in Appendix A. Those older adults interested in receiving more information told the nurses and those nurses notified the first researcher. After receiving further details, the older people who agreed to participate were subsequently contacted to arrange the interview at a place and time agreed upon via consensus. The site was, in all cases, the older person’s home, according to their wishes.

Regarding recruiting Spanish migrant nurses, a snowball sampling was followed, starting with a Spanish migrant nurse that the first researcher knew beforehand, knowing that she met the selection criteria.

### 2.5. Interview Guide

Concerning the interview guide, the first researcher prepared a preliminary script for each group of participants. These scripts had their pilot test with three older Norwegian patients and three Spanish immigrant nurses. We decided to do this to calibrate its content and assess whether we were approaching the interview correctly. These primitive interviews were not incorporated into the final corpus of the study. The reasons for their exclusion were, in the case of the older Norwegian patients, being slightly younger than 75, and in the case of Spanish migrant nurses, not having worked in home nursing services.

Once the final guide was calibrated, interviews were conducted, on the one hand, with older patients and, on the other, with Spanish migrant nurses (see Appendix A).

### 2.6. Data Collection

In the case of older Norwegian patients, we conducted eleven semi-structured face-to-face interviews, following the discourse saturation criterion [35]. To engage in culturally sensitive interview practices that would enrich the data and solve possible misinterpretations of the study phenomena in the specific cultural context of nursing home visits in older Norwegian patients, we focused on the cultural context and the interviewer–interviewee encounter [36]. This was possible through an assumption of sociocultural elements that shaped the meeting, such as the cultural identity of the interviewer and the older Norwegian patients, the space where the interviews were conducted (their homes) and the resulting intersection between all these elements [37]. These assumptions were reflected in the first researcher’s hermeneutical diary.

In the case of Spanish migrant nurses, we performed four semi-structured interviews via videoconference through the Zoom platform. To ensure optimal ethical and methodological considerations during the interview process with the Spanish migrant nurses, the first researcher followed the practical implications given by Pilbeam et al. [38]. We chose to use semi-structured interviews as a tool because it gave us some critical points to pivot for data collection; simultaneously, the depth of the participants’ discourses was not limited [39,40]. The interviews were conducted between November and December 2022.

Firstly, the interviews began with preliminary and open questions so the participants could respond freely and convey their first impressions. Subsequently, the semi-structured interview was progressively outlined and took a more defined orientation. All the interviews were audio-recorded and written in the hermeneutical diary, which the first researcher used for complementary information, such as self-hermeneutic considerations, kinesics, and body language [41,42].

We understood the proper language translation process as essential to accurately convey participant meanings between languages and ensure qualitative research’s reliability [43]. The interviews were conducted in Norwegian for the older Norwegian patients, and in Spanish for the Spanish migrant nurses, which was the original and native language of the participants. Hence, our translation process began in the dissemination phase of our work in English [44]. During this process, we translated the research, so we adopted the figure of researcher–translator. For this, the first researcher translated the speech acts of the older Norwegian patients based on his previous knowledge of the Norwegian language and the notes belonging to the hermeneutical diary. Regarding the Spanish language, the first researcher also translated the speech acts of the Spanish migrant nurses since Spanish was his mother tongue. In addition, the consultation technique was carried out with a native Norwegian nurse researcher, knowledgeable in both the language and the culture [45].

### 2.7. Data Analysis

For the analytical phase, we employed the hermeneutic analysis considerations described by Crist and Tanner [46] and linguistic analysis of the research corpus. Regarding CDA, linguistic analysis is necessary to decode the sociocognitive relationships hidden behind the discourse of older Norwegian patients and Spanish migrant nurses. However, we maintained sight of the fact that in the case of nursing inquiry, we must consider above all the study phenomena that interact in the context of care and the subsequent complementary linguistic component [47,48]. Through CDA, we identified discursive strategies that unravel the positive representation of the *self* (in-group favouritism semantic macro-strategy) and the negative representation of the *other* (out-group derogatory semantic macro-strategy) [49,50]. These discursive strategies representations were identified by analysing speech acts that illustrated comparisons, exemplifications, generalisations, polarisations, presuppositions and victimisations [50,51].

To carry out the approach to CDA, first, we had to find the moments that reflected situations of inequality from a critical point of view. To achieve this purpose, the research corpus was determined following the steps recommended by Bolívar [52]: (a) differentiating the textual corpus from the research corpus, (b) selecting informative material through an awareness of the basic assumptions and problems to be addressed, (c) determining the linguistic sublevels to be approached. To complete these phases, the first researcher conducted a first immersive reading of the transcripts to become familiar with the textual corpus (the participants’ speech acts). After that, a second comprehensive reading was carried out to identify the discursive moments that allowed us to resolve our objectives around active listening to preferences, shared decision-making and participation in care from a prism of inequality and power imbalance. Finally, a focused reading was carried out to understand the linguistic planes of the discourse that the corpus invited to address—since within the linguistic level, there are numerous sublevels— and thus decide the approach corresponding to the sharp fragments to analyse. After obtaining the research corpus, we chose to approach the analysis of the pragmatic, syntactic, semantic, rhetorical-stylistic and cognitive sublevels, as well as a description of the discursive strategies deployed by the participants.

### 2.8. Rigour and Quality Guarantee

For this study’s rigour and methodological quality, see Appendix A, where we detailed the guide for the reflexivity experience of conducting sensitive qualitative research as a cultural outsider, according to Joseph et al. [53].

### 2.9. Ethical and Legal Aspects of the Study

We informed all the participants about the characteristics of this study before the interview through an information sheet for each participant. Informed consent to participate in this study and audio recordings were obtained via signature. We previously told participants that the data collected would be used for research purposes only and that all identifying information would be anonymised to safeguard their identity. We informed the participants about (a) the objectives of the research, (b) the guarantee of the confidential nature of personal data, (c) the custody and handling of the data, (d) the disclosure of the results of this research and (e) the possibility of leaving the study at any time and without any consequences.

The present study was conducted in compliance with the principles of the Declaration of Helsinki and had permission from the Ethics Committee for the Province of Cordoba (Spain) (Minutes No. 283 Ref: 4118). Concerning Norway, we had approval from the Regional Committee for Medical and Health Research Ethics in Norway (Ref: 2018/1262, REK sør-øst) and the Norwegian Centre for Research Data. The personal data obtained have been processed following the General Data Protection Regulation EU/2016/679, of 27 April 2016, and the provisions of the Organic Law 3/2018, of 5 December, on the Personal Data Protection and Digital Rights Guarantee.

## 3. Results

### 3.1. Description of the Participants

As for the older people, all were Norwegian, of whom eight were women, and three were men, with a mean age of 83 (Table 2). The Spanish migrant nurses were all white Spanish women, with a mean age of 26 (Table 3). The corresponding sociodemographic information was collected through a self-filled registration before the interviews.

### 3.2. Conceptual Map for the Synthesis of the Results

Figure 1 illustrates how the discourses of older Norwegian patients and Spanish migrant nurses shape the perceived reality in the context of situations of active listening, shared decision-making and participation in care. On the other hand, it also visually shows the interaction between the migratory experience of the Spanish nurses and the acquisition of cultural competencies regarding the exercise of their nursing profession.

### 3.3. Narrative Development of the Results

The results of this study show explicit representations of power imbalance as speech acts belonging to the participants’ discourses. The older Norwegian patients adopted a role at times predominantly subservient about the study phenomenon, but with flashes, at other times, of personal agency and security concerning their desires and needs. On the other hand, the social representation of Spanish migrant nurses was more blurred, given their status as foreigners and the work context in which they found themselves. Spanish migrant nurses displayed empathic and holistic conceptions of care but were not always translated into a horizontal nursing exercise, sometimes influenced by their transcultural experience. According to the marked differences in their discourses, the older Norwegian patients and the Spanish migrant nurses led in communication interferences and affected the appearance of real decision-making situations.

The major themes were synthesised until a higher level of qualitative meaning was found. Then, we reached the following study patterns: Imbalance in communicative and personal perceptions, Power triangle “older patient-home nurse-institution”, and Cultural shock. Next, we developed the approach to CDA orderly through the study patterns, the major themes that integrate them and the speech acts to which reference is made.

To facilitate the identification and origin of the results, we present them in Appendix A in an organised manner in the form of major themes, minor themes, and the extracts of the speech acts that comprise them. To provide an overview of how the emergent CDA results were hierarchically distributed, we also calculated, in Appendix A, the magnitude of the derived findings [54].

#### 3.3.1. Pattern A: Imbalance in Communicative and Personal Perceptions

This study pattern collected the results in the participants’ discourses about the three study phenomena of nursing care we proposed in this work. There were communication barriers indicated by the discourse of the older Norwegian patients, which the Spanish migrant nurses also highlighted. However, migrant nurses were aware of the importance of maintaining therapeutic communication. Regarding decision-making, the situation was unbalanced to the extent that older patients declared that they had practically no decision-making moments. Spanish migrant nurses, for their part, made a distinction between moments, attitudes and personal characteristics that were vital to promoting decision-making among older adults. Perhaps within a more balanced relationship regarding participation, the older Norwegian patients changed their discourse towards a much more active attitude, something the migrant nurses also appreciated and clarified active participation as a crucial concept they tried to carry out whenever possible. The most used discursive strategies were those of exemplification and generalisation.

##### Major Theme 1: Interferences in Communication (1)–(22)

The discourse of the older Norwegian patients showed, in most cases, communicative difficulties whose origin lay in various situations. Their discourses were strongly random, depending on which nurse would visit them. Two different circumstances marked the modulation of the communicative moment; the first was the personal qualities of the professional who visited her, which invited them to communicate or not depending on whether the older Norwegian patients considered the professional’s attitude acceptable. The second circumstance included the responses received directly or indirectly by those nurses, which caused an inhibitory effect on therapeutic communication if they did not meet the needs of the patients. In any case, these circumstances converged in a feeling of inferiority or self-imposed contempt, something visible through a discourse full of negative pragmatic connotations, conveyed through speech acts with negative illocutionary force (“*There may be someone worse*”, “*I feel a bit guilty*”) or exemplifications with devaluation explanations about nursing praxis. Another reason that acted as a communicative barrier was their own personal qualities, and transitional moods, with the clear explicator that the older Norwegian people were aware of these same qualities and moods represented a barrier but apparently did not seem to do anything to remedy it. The speech act (13) was noteworthy, as it stood out from other more direct ones like the rest, since the implicature of loneliness slipped in here, possibly connecting with needing more time to talk with the nurses.

In the case of the Spanish migrant nurses, their discourse reflected an assumption of the importance of therapeutic communication with older patients. At the same time, they resorted to exemplification strategies to reinforce their arguments. The discourse of the migrant nurses also gave prominence to the personal qualities of, this time, the older Norwegian patients, so there was a certain parallelism with the older adults in the conception of personal qualities as a catalyst for communicative situations. Migrant nurses projected, through discursive strategies of intensification, that older Norwegian people notably conveyed their opinions and complaints, leaving the feeling that positive things were not considered in the same way. We highlighted the speech act (21), in which a migrant nurse left as a reflection of the implicature of what extent the fact of validating the preferences of which older adults in doubtful health contexts should prevail over optimal care. The older Norwegian patients demanded through their speech acts not to be listened to in situations of illness and request for help, in addition to suggesting specific states of need. On the other hand, the Spanish migrant nurses gave importance to the moments of active listening. Still, they mentioned elements that make it difficult, sometimes going so far as to reconsider the extent to which it is beneficial to validate preferences unconditionally.

##### Major Theme 2: Asymmetry in Decision-Making (23)–(39)

The discourse of the older Norwegian adults was overwhelmingly unanimous in not having the possibility to decide on anything in particular, despite declaring that they could do so. Within this reality, there was a division of opinions through polarised discourses, with a sector in favour of not deciding and another inclined to do so. In (30), the exemplification intensified the impossibility of deciding on something that the older Norwegian patient wanted; in (31), the use of an agency-intensifying adverb (“*strongly*”) made the speech act gain illocutionary force to make visible the willingness to reverse the situation; the use of direct speech acts by several patients implied that the nursing professionals had not even offered sensitive moments to the decision of the older patients. Conversational discursive strategies, such as silence or laughter, were components of significant pragmatic load in the discourse of the older Norwegian patients that added a feeling of insecurity and bewilderment in the face of the possibility raised.

Concerning Spanish migrant nurses, the discourse differs according to the type of older patients to be treated. The exemplifications served as a tool to clarify that it was only in some instances when older patients were not included in the decisions. On the other hand, the partially projected social representation corresponds to a role of power using a lexicon with pragmatic connotations (“*conflictive*”, “*bosses*”), but, at the same time, a lexicon and syntax constructions of an empathic nature (“*you have to ask them*”, “*I don’t try to force them but to convince them*”). That is why the power influence from the perspective of Spanish migrant nurses’ discourse is diffuse and unclear. On the other hand, the discursive resource of using the first-person plural pronoun reflected a group identity, through which migrant nurses felt they were spokespersons for their professional group. In addition, their discourse added empathic overtones as in (38), through which the migrant nurse considered herself an external factor by giving her opinion as an outsider narrator about the decision-making situation in the receiving country, something that reached another level in (39), in which the nurse went further and through a generalisation resource, commented that it was a usual dynamic to validate patient decisions even when they were not in their full faculties, something that has already appeared previously about active listening.

##### Major Theme 3: Heterogeneous Participation and Its Influential Elements (40)–(58)

The discourses of older Norwegian patients were heterogeneous regarding their participation in care. A minority declared not to participate, mainly because they were tired, as in (47), where the older Norwegian patient’s speech act was zigzagging and full of pauses, reflecting the disconcerting and fatigued state in which she found herself. On the other hand, most older patients showed signs of a participatory attitude through a discourse full of exemplifications related to the interventions carried out by nurses, balancing the power relationships regarding this matter. The discourses of the older Norwegian patients who did participate described situations of supervision of the material that the nurses would use, notices about any last-minute changes, cleaning of materials or feedback on when they were available to receive the visit. Especially notable was the speech act (46), through which the patient uses intensifying resources (“*absolutely*”) that added illocutionary force to the argument, in addition to an argumentative development that justified why she considered participating essential. In (48) and (49), the patient’s desire to participate stood out, and the helplessness of being prevented by the nurses. The use of syntax parallelisms (“*They don’t let me get up and cut bread; they don’t let me mop the floor*”) reinforced the patient’s discursive argument, in addition to being accompanied by explanations that indicate that nurses did not only not allow her to do certain things, but also the nursing care she received should not interfere with her desire to participate.

As regards incorporating older adults into participation in their care, the discourse of the Spanish migrant nurses was practically unanimous: they recognised the right to participate, stressed the importance of doing so, the positive effect it had on older patients’ health, and appreciated that, in general, in Norway, this level of participation in the older patients is taken into account. On the other hand, through exemplifications, the migrant nurses established differentiations about which moments were sensitive for participation, focusing more on self-care and hygiene tasks; they also stated that it depended to a large extent on the cooperative attitude of the older patients, something that could be a handicap but also remediable as they got to know each patient personally. At the same time, Spanish migrant nurses highlighted the importance of asking whenever possible whether the older patients or their families. A discordant note was (57), in whose speech act the migrant nurse resorted to a generalisation resource to ensure that older patients typically let themselves be led by nurses.

#### 3.3.2. Pattern B: Power Triangle “Older Patient–Home Nurse–Institution”

In study pattern B, we discovered that power relations were not only limited to the nurse–patient pairing but also transcended that link towards an institutional figure that exerted oppression on the working conditions of home nurses. For that reason, a “chain of power” was assembled between older patients, home nurses, and an institutional figure. The lack of collaborative organisation with the older patients and the lack of time and personnel were shared elements in the perception of older Norwegian patients and Spanish migrant nurses, in addition to describing situations of unmet needs and mistrust in the work environment. In this case, the migrant nurses referred in their discourses to the effect of cultural influence on said feelings. The predominant discursive strategies were those of exemplification, generalisation and comparison.

##### Major Theme 4: Lack of Organisation, Time and Staff (59)–(93)

The discourse of the older Norwegian patients reflected predominantly negative aspects regarding the organisation of the visits. Through abundant generalisation resources (“*They come when it is best for them to arrive*”, “*they come when they want*”, “*they don’t always come when they should*”), assumptions were generated that pointed towards home nurses as a precise authority figure, who did not consider older people as active members to take into account. This caused discomfort in older patients through exemplification resources, reinforcing their arguments by declaring that they have come to interrupt their daily routines because of the home nurses. It was worth noting the victimisation resource used by (64), with a robust lexicon of pragmatic connotations, going so far as to declare that she was subordinate to the nurses explicitly. From the perspective of the discourses of the older Norwegian patients, the care organisation kept almost exact parallelism with the time organisation and the lack of personnel they perceived. Almost unanimously, they used generalisation resources regarding the lack of time they believed nurses suffered, impacting the quality of care received. In (74), the enumeration resource intensified the work overload suffered by nurses, in addition to the implicature that he was not just another number for nurses but felt like one. The feeling of undervaluation is extensible among the discourses of the older patients, as seen in (78), a speech act that went further by classifying her need as unimportant, establishing a hierarchy of patients based on the activity that the nurses were going to do. It is worth highlighting the speech act (81) of great illocutionary strength, whose purpose was to make the addressee of her message aware of the precariousness in which the patient believed nurses lived, thus achieving an empathic perlocutionary effect.

Regarding the problem of coordinating visits, the Spanish migrant nurses asserted through their discourses that they tried to reconcile schedules with the preferences of the older adults. However, the conjunction “*but*”, present in the speech acts (84) and (85), invalidated this intention, projecting an image of frustration and limitation due to the workload suffered by the home nurses themselves. The social representation emitted in this case by migrant nurses was that of professionals who were aware of the preferences of their patients but who, through no fault of their own, were unable to satisfy them. Therefore, the power relations here extended beyond the nurse–patient pairing to end up in the nurse–institution pairing. The power triangle was then the subordination of the older patients to the home nurses, which followed the subordination of the home nurses to the institution that organised everything. This power triangle was directly connected with the perception of Spanish migrant nurses regarding the lack of personnel. This element saw a continuation concerning the perception of older Norwegian patients. The nurses’ discourse used numerous exemplification resources to intensify the lack of time that affected them.

On the other hand, in (88), a migrant nurse used a comparison resource to verify that, despite this, in her homeland, they had even less time. The nursing discourses empathised with the older patients. The Spanish migrant nurses declared they liked to spend time talking with the patients and that sometimes they could not do it—something that seemed *not the best thing to do*. Therefore, there were situations where care could be improved. In the speech act (93), an exemplification of the migrant nurse ventured to ensure that, in her current workplace, it was not so much a lack of time as a lack of self-organisation of the staff or even a lack of interest in certain colleagues in spending quality-care time with their older patients.

##### Major Theme 5: Consequences of Lack of Organisation, Time and Staff (94)–(116)

The lack of time and personnel on the part of home nurses resulted in the appearance of unsatisfied needs and unfulfilled wishes on the part of older Norwegian patients. The discourses of the older Norwegian patients used numerous exemplifications of the issues they considered necessary to address, which varied. The common denominator of the requests was to receive more help, in general, and more assistance regarding health procedures with others professionals and extra support at specific seasons. An issue that deserved to be highlighted was the speech act (95). An older patient declared that she would prefer to receive personalised-continuity assistance during the visits so that the nurses stayed the same continuously to avoid repeatedly explaining how they should act, which was *tiring*.

This problem was directly related to trust issues with home nurses, which surfaced in the discourse of older Norwegian patients. Employing exemplifications using the parallel syntax (“*Some don’t know precisely what to do* […], *some fool around a bit*…”), the discourses gave an account of the problem of the individual irregularity of nurses, which led to discontinuity of care. In the speech act (101), the first person plural was used, accompanied by a pragmatically charged lexicon (“*We, the dependents, must have confidence*”), to speak through a group identity about a great need that affected both himself and the group of older patients who received nursing services at home. At this time, the older Norwegian patient assumed a subordinate role that, from a sociocognitive point of view, outlined the dependency relationships he maintained with the home nurses. Another example of the power imbalance perceived by older patients could be found in (102), where the syntactic construction “*But I’m not in that position*” had the same semantic-cognitive sense of subordination. As a definitive cause of trust fluctuations, once again, we were able to find the assumption that, depending on the nurse, relationships will be more or less fluid, as observed in (105).

From the point of view of the discourse of Spanish migrant nurses, discursive strategies of exemplification, generalisation, and comparison were abundantly used to support their arguments. For the most part, migrant nurses considered that older Norwegian patients did not see all their needs always met, both from a physical-care and psychological–emotional point of views. The psychological component was of great importance in the nurses’ discourse, which was strongly affected by the lack of time to devote more attention to this plane and by perceived cultural components, which contrasted markedly with what home nurses perceived in their native country, as seen in the speech act (111). A discordant and explanatory note of the changing reality of home nursing perceived by Spanish migrant nurses was in the speech act (112), through which a migrant nurse declared that sometimes they performed services that were not necessary for older patients with the excuse of satisfying the need of the older adult’s family to keep an eye on their parents.

The general mistrust was also transmitted in the environment of migrant nurses, especially from a cultural point of view. In their discourses, exemplification and stylistic resources such as metaphors (“*other colleagues have let me pass the buck*”) were present, which endowed the arguments with illocutionary force. Entering a different work, culture and language scenario sometimes made them need more time to adapt to the environment and the patients. Added to this, they had to live with the presuppositions of other Norwegian colleagues who considered Spanish migrant nurses to be more skilled and, therefore, were burdened with more laborious tasks or tasks that did not correspond to them, in addition to dealing with colleagues who did not generate confidence when realising their tasks, which was why the feeling of insecurity was present in the Spanish migrant nurses’ discourse on certain occasions.

#### 3.3.3. Pattern C: Cultural Shock

Study pattern C allowed us to appreciate all the effects of culture shock on the perception and relationship with the work environment of Spanish migrant nurses. From a personal point of view, it represented a challenge to adapt to an environment culturally so different from the one they came from, even going so far as to reconsider their plans for the future after experiencing nursing in Norway. From a labour point of view, migrant nurses´ discourses showed moments of harmful discrimination by patients and colleagues and moments of positive discrimination that, in the end, turned against them for their nursing tasks. The most used discursive strategies were those of exemplification, comparison and polarisation.

##### Major Theme 6: Cultural Issues in the Migration Experience (116)–(136)

Numerous exemplifications enriched the discourse of Spanish migrant nurses regarding their personal experiences in their labour journey through Norway. The situational and personal context marked these experiences, and the migrant nurses highlighted the positive and negative aspects of living in Norway through comparative strategies. Their discourses were full of references to Spain. At the same time, they highlighted the difficulty of adapting to a climate, environment and culture so different from that of their homeland. Regarding the personal perception of working conditions in Norway, we considered it interesting to highlight the speech act (119), in which the resource of syntax parallelism is used (“*I don’t want to be called to work for two months, my contract runs out; they call me for another month, and my contract runs out*”) to polarise the perceived job insecurity in Spain compared to Norway. Prejudices also appeared in (122), in which a migrant nurse assumed that her assumption about the Norwegian people was wrong, which was less favourable than what she experienced.

Regarding the cultural experience of Spanish migrant nurses in their jobs, an adverse circumstance linked to perceived moments of racism abounded in their discourses. In this sense, the lexicon used was quite explicit and contained a tremendous illocutionary force (“*I felt attacked*”, “*they have had some racist problems*”, “*because you are a foreigner*”, “*I have felt somewhat discriminated against*”), which conveyed a solid discriminatory feeling in the workplace, something that made them adopt an unbalanced relationship of power with their Norwegian peers. The work role of the migrant nurses was the same as that of their Norwegian counterparts. Still, due to occasional discriminatory circumstances, they were viewed as somehow inferior by their peers and by the older Norwegian patients themselves, as evidenced by speech acts (123), (124), (130)–(132) and (134). On the other hand, the discourse of migrant nurses also reflected culturally different perceptions regarding colleagues and patients, which disoriented them a bit, as reflected in the speech act (125) or (135). Returning to the situations of discrimination, it was noticeable that they suffered not only moments of harmful discrimination but also positive discrimination. In other words, and in line with (126), (128) and (133), the Spanish migrant nurses were aware, through generalisations, that their work and training were reasonably well regarded in Norwegian nursing. However, concerning this positive discrimination, through exemplification strategies and their use of lexicon (“*I may have felt a little forced to do something in some situations*”), we could interpret in their speech acts that it was something that did not seem entirely appropriate to the Spanish migrant nurses since they were made to take care of specific tasks that did not belong to them or that they were not previously consulted.

## 4. Discussion

The situations of active listening, decision-making and participation in care were the epicentre of the imbalance in power relations between older patients and home nurses. The importance of reconciling active listening, decision-making and participation in older adults has already been highlighted by Koskenniemi et al. [55], who defined acceptance, active listening, commitment and warmth as pillars of care based on respect [56]. However, our results highlighted barriers to the appearance of active listening situations due to introverted personal characteristics from the older patients´ side or less empathic responses from the home nurses´ side. Nevertheless, this contrasted with the discourse of the Spanish migrant nurses, who claimed to make great efforts to dedicate time to therapeutic communication since they considered it essential to listen to older people, something shared by various studies that underline the importance of inclining to listen to older patients and take their preferences seriously [57,58]. On the other hand, in the discourse of the migrant nurses, there was a component of helplessness due to the lack of time that would allow them to develop easier active listening towards all older patients, which was consistent with the findings of Anshasi et al. [59].

Regarding shared decision-making situations, the results of this study revealed a bleak picture due to the non-existence of these in the case of older Norwegian patients. These results are consistent with the findings of a recent study conducted by Keij et al. [60]. With a very similar sample size, older patients found it more challenging to participate in shared decision-making because they were victims of a paternalistic decision-making style that prevented them from absorbing information and expressing themselves. However, according to the migrant nurses’ discourse, these situations were influenced by the attitude of the older adults. This apparent contradiction makes sense when related to other studies that consider this process a phenomenon full of uncertainty, one that requires a dialogue attitude on the part of the older patients and health personnel [61,62]. There was more uniformity in the discourses regarding active participation since most of the older patients stated that they participated in tasks related to their care. The migrant nurses extolled the vital role that introducing them had for older patients’ health, which agrees with other studies that understand the participation process as an act that requires equality in power relations and communicative understanding [63,64].

Regarding the consequences of this imbalance in power relations, the discourse of the older Norwegian patients and the Spanish migrant nurses converged on the adverse effects of a deficient care organisation and the professionals’ lack of time. In addition, older people and migrant nurses showed signs of mistrust towards the work environment due to their perceptions about the sometimes-inadequate training of some colleagues. All this related to the organisation of care, lack of time and distrust agrees with the study findings obtained by Bravell et al. [65] in a Nordic work environment, which is culturally transferable to the present study.

Regarding the migratory experience, this study has emphasised the effect of culture shock that Spanish migrant nurses have experienced in the care they provide. The results of this study show, broadly, culturally competent migrant nurses regarding Campinha-Bacote’s Model of Care of Cultural Competence in the Delivery of Healthcare Services [27,66]. Spanish migrant nurses were aware of the images they carried about the Norwegian people, something they identified in their migrant experience as nurses and discussed this experience with the researcher, which is consistent with the concept of cultural awareness [67]. In addition, through their speech acts, we could ascertain that Spanish migrant nurses could assess the older Norwegian patients in their cultural context, even adapting at specific times to individual wishes or needs. This meets the cultural skill concept. As for the concept of cultural encounters, the migrant nurses fulfilled it satisfactorily since not only did they not shy away from face-to-face meetings with older Norwegian patients despite specific language restrictions, but the nurses also sought out these encounters to establish communication and get to know their patients. The last concept of cultural desire represents the cornerstone of cultural competence since it feeds energy and motivation to the health professional to become culturally competent [68]. This was something we discovered throughout the discourses of the Spanish migrant nurses, as a result of this study, despite sometimes suffering work overload that prevented them from carrying out their tasks satisfactorily.

The labour discrimination perceived through the discourses of the migrant nurses revealed that the lack of proficiency in the Norwegian language acted as a limiting element in the relationship with other colleagues or patients, who even went so far as to demand the presence of Norwegian nurses, something that agreed with studies exploring the experiences of other migrant nurses in Norway [69,70]. However, despite classifying these as racist experiences, the Spanish migrant nurses declared that they also perceived positive discrimination at the same time as they were considered skilled and highly valued, consistent with the work by Munkejord [71]. On the other hand, better labour and economic conditions were one of the reasons why migrant nurses leapt abroad to work as nurses. This made them reluctant to return to their homeland in the short term following their speech acts, thus agreeing with other studies about migrant nurses in Norway [69,72]. At the same time, these nursing migration results in the Norwegian context are also consistent in non-European settings, such as the United States as a country suffering from a nurse shortage [73]. This deficiency is partially resolved by welcoming nurses of foreign origins, such as those from the Philippines, whose reasons for taking the step to undertake their profession outside their native country are partly economic and labour conditions, as our results also reflect with our study participants [74].

The use of specific discursive strategies and arguments made the older Norwegian patients and the migrant Spanish nurses project different social representations. Older Norwegian patients generally adopted a double image, as a victim, on the one hand, for not feeling involved in their care at times and for not being treated as they deserved by home nurses; of agency, on the other hand, through a discourse that sometimes made explicit the desire to change things. Spanish migrant nurses adopted, for their part, an image committed to the profession and the care they professed, but at the same time, said care was influenced by negative experiences at times, from a cultural and organisational point of view, in addition to pointing out also to older people sometimes being responsible for frustrated moments of active listening, decision-making and participation. This coincides with Tajfel’s theory of social identity [49,75], which understood the attitude of the members of different social groups as a reflection of the identity of belonging to each group, whose purpose is to positively visualise their actions above that of the others, on which negative aspects stand out [51].

### 4.1. Strengths and Limitations of the Study

The main strength of this study lies in incorporating the sociocritical perspective of DS in the context of the migrant experience of Spanish nurses. The CDA has already been shown to be helpful in nursing research, as proposed by Powers [47]. This study also makes visible a reality that is increasingly present in our days, in which the current demands in the healthcare system worldwide push many nurses to decide to emigrate for leaving socioeconomic conditions that do not fit with the training and work capacity that they acquired in their countries of origin. On the other hand, we have invested a great effort in describing in detail the methodological framework of this study, in addition to adequately organising the results obtained, providing detailed information on the preparation of the research and following rigour and quality criteria following the transcultural perspective that we have adopted.

We decided not to have older patients who suffered from cognitive impairment or were terminally ill. This could be considered a limitation, but we understood these subgroups of older patients as another different patient profile requiring particular care, which somehow influences the discourse. On the other hand, the short number of Spanish migrant nurses participating in this research can be considered limiting. However, it was already difficult to find Spanish migrant nurses who met the agreed selection criteria. On the other hand, we refer to Malterud et al. [76], who state that in qualitative research, a sample rich in information comes from a low number of participants who confer it. Another area for improvement of this study is that it was not possible to triangulate the transcribed material with the participants after the data collection, despite performing an intra-interview triangulation.

### 4.2. Relevance to Clinical Practice and Healthcare Policies

The present study has explored power relations between older Norwegian patients and Spanish migrant nurses. The negative effect of a specific imbalance on situations of active listening to preferences and shared decision-making reveals the urgent need to develop measures and interventions that promote a horizontal relationship, in terms of collaboration, between nurses who provide services at home and older patients who receive it. It is vital to implement alternative models of response to the health problems of older adults in the community so that it is possible to include them in the decision-making processes to provide quality nursing care [77]. On the other hand, a detrimental context in which home nurses find themselves has been revealed, which causes an overload of work, deficiencies in the care organisation and mistrust in the work environment, the effects of which are well-known and endorsed by the scientific literature, harming the quality of care. For this reason, it becomes urgent to listen to and attend to the needs of home nurses and also to attend to the power imbalance between home nursing staff and home-care management organisations. Finally, we want to highlight the importance of caring for migrant nurses, who deserve to be considered given their culturally unique condition, so that they feel inserted and move away from discriminatory experiences to confer added value to their transcultural care so that special attention from health systems and agencies is necessary.

## 5. Conclusions

This study has revealed an imbalance in power relations between older Norwegian patients and Spanish migrant nurses in in-home nursing. Active listening situations were influenced by elements such as limiting personal qualities of older Norwegian people and less empathic responses by home nurses in general, based on the discourse of older patients. The discourses of the Spanish migrant nurses coincided with the older adults in the personal aspect of these patients as a communicative barrier. Regarding decision-making situations, the discourse of the older people made it clear that at no time had they enjoyed the opportunity to decide, some older people being prone to have an attitude of agency concerning the process, and others a passive attitude. The discourse of the migrant nurses made it clear that they were aware of the importance of deciding together, but sometimes they seemed not to carry it out. Regarding participation in care, the discourse of the older patients and the migrant nurses were more in unison, highlighting a participative intention in most older adults and a respect for engaging them from the home nurses. However, the true origin of this imbalance is mainly affected by the institutions that organise home care. The work context, the lack of organisation of visiting hours with the older patients, the lack of time and the distrust towards some members of the nursing team were an indirect consequence of a silent power relationship between the care institution and migrant nurses, the latter subjected to the former. Finally, the culture shock had a significant positive and negative weight on how Spanish migrant nurses related to their environment, thus shaping their perception and way of providing nursing care. Regarding the latter, the speech acts of Spanish migrant nurses have revealed that they are culturally competent professionals. The social representations emitted by the older Norwegian patients were victims for being the last link in power relations; on the other hand, patients wanted to be treated with dignity and not have assumptions made about what they want and need. The social representations of Spanish migrant nurses were those of a professional group committed to their work, critical of specific situations of social injustice and influenced by cultural elements in their migratory experience.

## Figures and Tables

**Figure 1 healthcare-11-01282-f001:**
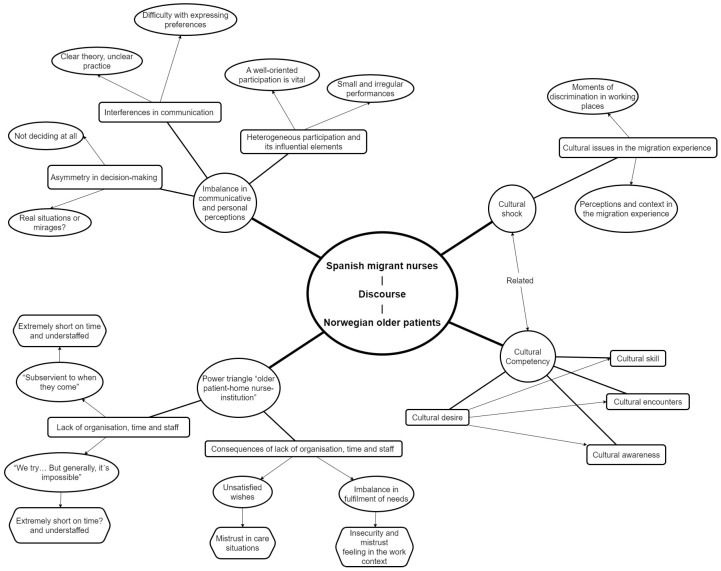
Discursive and power elements shaping relations between older Norwegian patients and Spanish migrant nurses.

**Table 1 healthcare-11-01282-t001:** Selection criteria for older Norwegian adults and Spanish migrant nurses.

	Older Norwegian Adults	Spanish Migrant Nurses
Inclusion criteria	**A**. 75 years old or older. **B**. Living alone at home. **C**. Receiving nursing home care services at the time of the study.	**F**. Having at least six months of work as a nurse in Norway. **G**. Having at least one month of uninterrupted work experience as a community care nurse in Norway. **H**. Having performed nursing home care visits to older Norwegian patients as part of the community care nursing tasks.
Exclusion criteria	**D**. Suffering from cognitive impairment. **E**. Suffering from a terminal illness.	**I**. Not having made a minimum of one nursing home care visit to older Norwegian patients in at least the one month worked as a community care nurse.

**Table 2 healthcare-11-01282-t002:** Characteristics of older Norwegian patients (*n* = 11).

Participant No.	Age	Sex	Home Nursing Services (Frequency)	Older Patients’ Health Condition	Home Setting (Urban/Rural)
1	86	Woman	Medicine delivery (three times a day) Administration of nebulisers (four times a day) Placement of compression stockings (once a day) Removal of compression stockings (once a day) Shower aid (once a week)	COPD Hypertension Peripheral Vascular Disease	Urban
2	89	Woman	Medicine delivery (three times a day) Shower aid (once a week)	Hypertension Hyperlipidaemia	Urban
3	87	Woman	Medicine delivery (once a week)	Hyperlipidaemia	Urban
4	88	Woman	Medicine delivery (three times a day) Administration of nebulisers (twice a day) Application of analgesic ointment for pain (if necessary)	Asthma Arthritis	Urban
5	81	Man	Medicine delivery (once a week) Irrigation for urinary catheter maintenance (twice a week)	Prostatic hyperplasia	Rural
6	91	Woman	Medicine delivery (once a week) Placement of compression stockings (once a day) Removal of compression stockings (once a day) Medication intake check (once a day)	Peripheral Vascular Disease Hyperlipidaemia	Urban
7	75	Woman	Medicine delivery (once a day) Administration of nebulisers (twice a day) Lunch delivery (once a day)	COPD Smoker Coagulation disorders Malnutrition (deficit)	Urban
8	75	Man	Medicine delivery (once a week) Medication intake check (twice a day) Food intake check (twice a day)	Insomnia Anxiety Malnutrition (deficit)	Urban
9	83	Woman	Placement of compression stockings (once a day) Removal of compression stockings (once a day) Moisturising foot cream application (once a day)	Peripheral Vascular Disease	Rural
10	78	Man	Placement of compression stockings (once a day) Removal of compression stockings (once a day)	Peripheral Vascular Disease	Rural
11	79	Woman	Blood glucose levels check (twice a day) Insulin administration (if necessary)	Diabetes Malnutrition (excess)	Rural

**Table 3 healthcare-11-01282-t003:** Characteristics of Spanish migrant nurses (*n* = 4).

Participant No.	Age (Years)	Qualified Nursing Professional (Years with/or Months)	Nursing Professional Employment (Years with/without Months)	Nursing Professional Employment in Norway (Years with/without Months)	Nursing Professional Employment in Home Care Services (Years with/without Months)	Workplace in Norway
12	28	6 years	4 years and a half	4 years and a half	4 years	Råde
13	25	3 years	2 years and a half	2 years and a half	1 month	Odda
14	26	3 years	3 years	2 years	1 year and 5 months	Høyanger-Odda
15	26	1 year and 7 months	7 months	7 months	3 months	Sola

## Data Availability

Not applicable.

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
