# Peer review of "Analysing Power Relations among Older Norwegian Patients and Spanish Migrant Nurses in Home Nursing Care: A Critical Discourse Analysis Approach from a Transcultural Perspective"

_healthcare, 2023, doi:10.3390/healthcare11091282_

Round 1

Reviewer 1 Report

The findings of this study provide valuable insights into cross-cultural elder care issues. However, there are some shortcomings that need improvement:

1.Line 43, it is recommended to use a decimal point instead of a comma to indicate the percentage (92.84% instead of 92,84%).

2. In Table 1, the inclusion criteria define the age as 65+, but it is unclear why the study used 75+ years old.

3. In Table 2, it would be useful to provide information on the duration of nursing home care services received by the older adults, as well as their exclusion scores for cognitive impairment (e.g. MMSE, AD8). The study duration should also be clearly stated (e.g. from 2022 until present). Additionally, it would be beneficial to include more details about the participants, such as their ageism, economic status, and co-morbidities.

4. Line 162 mentions Scheme S2a & S2b, but the appendix refers to them as S3a and S3b.

5.To ensure clarity, it would be helpful to provide the translated English version of "230301_Scheme-S1-booklet-norwegian-patients" for readers.

6.The components in Figure 1 were difficult to read and should be made more clear.

7. The study findings are based on a small sample size of 11 participants and, thus, caution is advised when drawing conclusions. The discussion should also include a comparison of similar quantitative research findings from other studies to support or reject the points made. Additionally, it would be useful to explore similar situations in other countries, such as older adults in the US and nurses in Indonesia.

Author Response

Dear Reviewer 1,

I am attaching the WORD file we created with all our responses to your comments.

We are grateful for your interest and time invested in reviewing our manuscript. We firmly believe that our manuscript has been improved thanks to your suggestions.

We sincerely hope that you find our responses enlightening.

Thank you again,

The research team.

Reviewer 2 Report

This is an excelent article, very original and presenting useful results. 

I suggest you provide references or further explanation to this:

"the positive representation of the self (in-group favouritism semantic macro-strategy) and the negative representation of the other (out-group derogatory semantic macro-strategy)." (l. 208-210)

Clarify this (I guess you refer to first-person plural pronoun):

"On the other hand, the discursive resource of using the first-person pronoun reflected a group identity, through which migrant nurses felt they were spokespersons for their professional group" (l. 361-363)

And proofread for minor writing mistakes.

Author Response

Dear Reviewer 2,

I am attaching the WORD file we created with all our responses to your comments.

We are grateful for your interest and time invested in reviewing our manuscript. We firmly believe that our manuscript has been improved thanks to your suggestions.

We sincerely hope that you find our responses enlightening.

Thank you again,

The research team.

Reviewer 3 Report

Dear Authors,

I found the article interesting, and the idea of using a critical discourse analysis framework with a transcultural perspective is consistent with the aims.

I would like to share some comments to facilitate a manuscript revision and increase reporting.

Abstract: You should state why power relations can be problematic between Norwegian older people and Spanish migrant nurses and what are power relations

Introduction:

1. the idea of the first paragraph, “nursing was one of the occupations with the most difficulty filling positions, and whose temporary rate in labor contracts was 92,84% “ is very strange and requires to be contextualized (in addition, the link of the reference 2 does not work; I suggest a strong reference for this point). As per the last reports of the OECD, “Nurses in Spain increased to 6.10 per 1000 people in 2019 from 6.08 per 1000 people in 2018”, and this is in a world with a severe nursing shortage. My advice is to contextualize the context of Spain as a net supplier of nurses to foreign countries.

2. In addition, on page 2, lines 67-71, the concepts of hierarchies, power relations, and empowerment seem to be central and remain undefined. Please define and expand the meaning of the central phenomenon of “power relations” because keeping it undefined might generate ambiguity.  

3. The phenomenon of “Spanish migrant nurses to Norwegian older people in the community in the face of a possible imbalance of power and a consequent situation of inequality, discrimination or social injustice” seems to arrive “out of the blue”. It should be explained with background information. Has been this phenomenon already described? Otherwise, it is not clear why authors identified this aspect as problematic if there is no link to previous descriptions or objective facts.  

4. Key concepts covered in the discussion and present in the title are missing in the introduction (active listening, decision making, and participation)

Methods and results: These sections are well described.

Discussion: Some ideas presented in the discussion might not be fully supported by data or aligned with what the methods allow to determine. For example, in the first paragraph, it was reported that  “Norwegian older patients highlighted the rare occurrence of moments of active listening due to introverted personal characteristics or less empathic responses from home nurses. This contrasted with the discourse of migrant 564 Spanish nurses, […]”. Here the experience of 11 older Norwegian patients is key to contextualizing the results that cannot be intended in a whatever form of “inference” of the emerging perceptions due to to the nature of the employed methods. In other words, some statements should be more clearly refocused to avoid misunderstandings of the results (especially in the first paragraph and should be divided into at least two different paragraphs). 

Author Response

Dear Reviewer 3,

I am attaching the WORD file we created with all our responses to your comments.

We are grateful for your interest and time invested in reviewing our manuscript. We firmly believe that our manuscript has been improved thanks to your suggestions.

We sincerely hope that you find our responses enlightening.

Thank you again,

The research team.

Round 2

Reviewer 3 Report

Most of my earlier points have been handled. I was unable to thoroughly evaluate table 2 and figure 1 in the present version. Please format the manuscript in a way that makes it possible to evaluate Figure 1 and table 2 as well. 

Author Response

Dear Reviewer #3,

Regarding your last comment, we have indeed realised that Table 2 needed a readjustment to identify all its parts in the manuscript. We apologise for this, and we appreciate you notifying us about it.

We have readjusted Table 2 (page 6-7 of 22) so that all its columns and the new one we inserted, as we already said in our previous answers, are visible.

Figure 1 (page 9 of 22) did not appear in the PDF version of the revised manuscript due to some electronic error. It did appear in the WORD version, so we have formatted the Figure file in another extension so that it appears now in a central place on its respective page, both in the WORD version and the PDF version, making it possible to appreciate all its elements.

We hope that these latest modifications are sufficient for correct viewing.

Thank you very much for everything; we apologise for the inconvenience and look forward to your final decision soon.
